# Association between Stress Hyperglycemia and Adverse Outcomes in Children Visiting the Pediatric Emergency Department

**DOI:** 10.3390/children9040505

**Published:** 2022-04-02

**Authors:** Woori Bae, Moon Bae Ahn

**Affiliations:** 1Department of Emergency Medicine, College of Medicine, The Catholic University of Korea, Seoul 06591, Korea; baewool7777@hanmail.net; 2Department of Pediatrics, College of Medicine, The Catholic University of Korea, Seoul 06591, Korea

**Keywords:** stress hyperglycemia, child, emergency department, mortality

## Abstract

Stress hyperglycemia (SH) is often identified in patients visiting the pediatric emergency department (PED), and SH in adults has been associated with adverse outcomes, including mortality. In this retrospective study, we determined the adverse outcomes according to blood glucose (BG) levels of children visiting the PED of tertiary hospitals. Data were collected from the electronic medical records of children aged <18 years between 1 January 2011 and 31 December 2020. A total of 44,905 visits were included in the analysis. SH was identified in 1506 patients, with an incidence rate of 3.4%. Compared to those without SH, patients with SH had significantly higher ward admission rates (52.6% vs. 35.9%, *p* < 0.001), intensive care unit admission rates (2.6% vs. 0.7%, *p* < 0.001), and mortality rates (2.7% vs. 0.3%, *p* < 0.001). Compared to the normoglycemic group of 45 ≤ BG < 150 mg/dL, the odds ratios (95% CI) for mortality were 5.61 (3.35–9.37), 27.96 (14.95–52.26), 44.22 (17.03–114.82), and 39.94 (16.31–97.81) for levels 150 ≤ BG < 200, 200 ≤ BG < 250, 250 ≤ BG < 300 and ≥300 mg/dL, respectively. This suggests that SH is common in children visiting the PED and is associated with higher adverse outcomes. Thus, there is a need to quickly identify its cause and take prompt intervention to resolve it.

## 1. Introduction

Stress hyperglycemia (SH) is a transient elevation to 150 mg/dL or more in blood glucose (BG) levels during an acute illness, usually in people with no history of diabetes [1]. However, SH is also known to improve as the patient recovers [2,3]. There are numerous conditions associated with SH in children who visit the pediatric emergency room. According to Levmore-Tamir et al. [4], the most common diseases in children with SH at the pediatric emergency department (PED) visits were respiratory, neurologic, and gastrointestinal diseases. In addition, according to Valerio et al. [5], febrile seizure was the most common comorbidity in the presence of SH in children visiting the PED. This study also reported a three-fold higher rate of SH in children with febrile seizure compared to other febrile disorders. Furthermore, upon a child’s visit to the PED, it is critical to promptly administer certain drugs, such as inhaled beta-adrenergic stimulants, glucocorticoid, intravenous dextrose fluid, phenytoin, and antipyretics (acetaminophen or ibuprofen), which, at the same time, could lead to drug-induced hyperglycemia [6].

In patients with acute illness, when the metabolic demand increases, blood glucose rises in order to supply adequate fuel to vital organs, and this elevated blood glucose is called SH. The most common mechanism of SH is explained by increased gluconeogenesis and glycogenolysis, as well as insulin resistance stimulated by proinflammatory cytokines and counter-regulatory hormones secreted during critically ill conditions. This then leads to glycolysis and oxidative phosphorylation, resulting in excessive reactive oxygen species, cellular apoptosis, and multiple organ damage [2].

In most cases, the management of patients with SH in the PED requires no additional treatment or evaluation due to the absence of clear management guidelines. However, studies have reported that hyperglycemia is associated with adverse patient outcomes. One study reported that among pediatric patients who visited the PED for trauma the mortality rate of those with hyperglycemia of ≥200 mg/dL was 5.6%, which was significantly higher than those without SH with a rate of 0.6%. In addition, the hospital LOS in patients with SH was 16.4 days, which was significantly longer than the 7.8 days in patients with normoglycemia. Also, the ICU admission rate was 55.6% in patients with SH, which was significantly higher than the 20.9% rate in patients with normoglycemia [7]. In another hyperglycemia study, critically ill children who visited the PED had a mortality rate of 15.4%, while the mortality rate of those with normoglycemia was 8.0%, indicating a significant difference [8]. 

Therefore, this study aimed to identify the adverse outcomes in patients who visited the PED with and without SH. In addition, the trends of the adverse outcomes according to the increase in BG levels were also evaluated.

## 2. Materials and Methods

### 2.1. Study Setting, Design, and Population

We conducted a retrospective observational study of visits to the PED of a single tertiary hospital between 1 January 2011 and 31 December 2020. We included patients aged <18 years in this study. Patients without blood tests, those not assigned with a diagnostic code, and those who were diagnosed with diabetes or hypoglycemia were excluded from this study. The average annual number of PED visits was approximately 14,700 per year during the study period (range: 7241–17,539). Diagnoses were made using the diagnostic codes of the Korean Standard Classification of Diseases-7 (KCD-7) based on the International Classification of Diseases-10 (ICD-10) of the World Health Organization. Each PED visit was considered as an independent case from PED arrival to discharge. The following variables were extracted from the electronic medical records of the hospital for all eligible visits during the study period: age, sex, laboratory blood glucose (BG) levels, vital signs at presentation, mode of arrival, diagnoses made in the PED, PED length of stay, administration of vasopressors in the PED, ward admission, pediatric intensive care unit (PICU) admission, and mortality. Diagnoses were classified into nine categories: respiratory, gastrointestinal, hemato-oncologic, neurologic, trauma, infectious, urogenital, and cardiac disorders. Vasopressor administration was defined as the administration of one or more vasopressors in the PED. SH was defined as hyperglycemia with a BG level of >150 mg/dL (8.3 mmol/L) occurring in non-diabetic patients with acute illness [9]. Hypoglycemia was defined as a BG level of <45 mg/dL (2.5 mmol/L) [10].

### 2.2. Outcomes

We evaluated the mortality rate, ward admission rate, vasopressor administration in the PED, and PICU admission rate in patients with SH and normoglycemia at the time of the PED visit. We also evaluated the mortality rate, ward admission rate, administration of vasopressors, and PICU admission rate according to the patient’s BG level at the time of the PED visit.

### 2.3. Statistical Analysis

Statistical differences between the two groups were analyzed using the Chi-squared test for categorical variables and the Mann–Whitney test for continuous variables. The BG level was used as the independent variable, while ward admission rates, vasopressor administration in the PED, PICU admission, and mortality were used as the dependent variables. For the ward admission rate, administration of vasopressors in the PED, PICU admission rate, and mortality rate, the odds ratio (OR) according to BG levels was calculated at a 95% confidence interval (CI) through binary logistic regression analysis that was adjusted for age, sex, diagnosis, and mode of arrival. All statistical analyses were performed using R version 4.0.0 (R Foundation for Statistical Computing, Vienna, Austria), with the probability level for significance set at *p* < 0.05.

## 3. Results

From January 2011 to December 2020, a total of 146,836 patients visited the PED of a single tertiary hospital. Among them, there were 90,843 patients who did not undergo blood testing and 10,574 patients who were not assigned with diagnostic codes, which were excluded. In addition, 346 and 168 individuals with diabetes-related diagnoses and hypoglycemia, respectively, were also excluded. For this study, 44,905 patients were finally included (Figure 1). The number of patients in the SH group was 1506 (3.4%), with a mean age of 75 ± 65 months, which was significantly lower than the mean age of 83 ± 68 months (*p* < 0.001) in the normoglycemic group. The proportion of the 1–3-year-old group was higher, while the proportion of the 7–12-year-old group and 13–17-year-old group was lower in the SH group than in the normoglycemic group. The proportion of the female sex in the SH group was also significantly lower (44.8% vs. 41.2%, *p* < 0.001). The diagnoses that are most common in the SH group, starting with the highest, are, in order, respiratory, gastrointestinal, and hemato-oncological diseases. Meanwhile, for the normoglycemic group, the most common diagnoses from highest to lowest, are gastrointestinal, respiratory, and hemato-oncological diseases. The demographic data of the patients in the study group are summarized in Table 1.

The PED mean length of stay (LOS) of the SH group was 9.2 ± 10.9 h, which was significantly longer than that of the normoglycemic group, with a mean LOS of 7.4 ± 9.0 h (*p* < 0.001). The ward, vasopressor, and PICU admission rates of the SH group were 52.6%, 5.1%, and 2.6% respectively, which are all significantly higher than that of the normoglycemic group with rates of 35.9%, 0.5%, 0.7% (*p* < 0.001), respectively. Furthermore, the mortality for the SH group was significantly higher than that of the normoglycemic group, with a rate of 2.7% compared to 0.3% (*p* < 0.001), as summarized in Table 2.

To study the clinical outcomes according to BG levels, the patients were divided into the following five BG groups: normoglycemic group, 45 ≤ BG < 150 mg/dL; 150 ≤ BG < 200 mg/dL; 200 ≤ BG < 250 mg/dL; 250 ≤ BG < 300 mg/dL; and ≥300 mg/dL. The ward, vasopressor, and PICU admission rates in the PED all increased significantly, as presented in Figure 2. For each BG group, the ward admission rates were 35.9%, 48.9%, 65.3%, 67.9%, and 80.4% (*p* < 0.001), respectively. The vasopressor administration rates were 0.5%, 3.2%, 9.7%, 16.1%, and 23.5% (*p* < 0.001), respectively. The PICU admission rates were 0.7%, 1.6%, 5.1%, 8.9%, and 9.8% (*p* < 0.001), respectively. Lastly, the mortality rate also significantly increased to 0.3%, 1.4%, 6.8%, 8.9%, and 11.8% (*p* < 0.001) for each BG group, respectively.

Compared with the normoglycemic group, the OR (95% CI) for adverse outcomes including hospitalization, vasopressor administration, PICU admission, and mortality for each group of SH are specified in Table 3.

## 4. Discussion

For critically ill patients visiting the PED, measuring the BG level is a fundamental assessment process within a routine care setting. Capillary BG is a rapid and easy measure performed on patients upon arrival in the PED and this correlates well with venous-derived measurements [11]. In our study, we found that the degree of SH was proportional to the severity of patient outcomes, which was also reported in an Israeli study [4]. Although the level of glucose elevation induced by stress may vary depending on the type and severity of the disease, a significant relationship between SH and outcome parameters may provide important clinical clues for disease prognosis. In terms of mortality rate in particular, our results were consistent with findings from previous studies conducted in adult populations. Cinar et al. reported the mortality rate among patients diagnosed with acute myocardial infarction was approximately 10 times higher in SH groups than non-SH groups [12]. In another study by Fabbri et al., a high stress hyperglycemia ratio among all outcome predictors increased the risk of mortality by over five times in elderly diabetic patients following diagnosis of sepsis-related hospitalization [13]. Likewise, the mortality rate in our results revealed a 44-fold increase in SH populations compared to normoglycemic populations. To the best of our knowledge, this is the first Korean study to analyze data on the adverse outcomes of SH in pediatric patients visiting the PED.

Interestingly, the frequency for children aged 1–3 years was the highest in the SH population compared to other age groups, and the second highest among the entire PED visitors. In our study, patients were subcategorized by age and disease type, not by chief complaint, and the most common reason for PED visit among children aged 1–3 years was febrile illness. Higher frequency of febrile populations might have affected the SH incidence of children aged 1–3 years. The study by Levmore-Tamir et al. (2020) also supports our findings in which SH incidence appeared to be highest in the respiratory tract, followed by gastrointestinal diseases, which were the two most common major diagnoses [4]. Compared to patients with normoglycemia, children with SH were more likely to be at risk of poor outcomes, such as PICU admission and mortality.

In Figure 2, all outcome parameters had a tendency to increase as the BG levels increased. Likewise, all adjusted odds ratios for the patient outcomes appeared to be the highest in the extremely hyperglycemic (BG > 300 mg/dL) group except for PICU admission and mortality (Table 3). Possible explanations are (i) statistical limitations due to the small sample size of the extremely hyperglycemic group, and (ii) exclusion for those numbers who expired in the PED unit could have affected the odds ratio for PICU admission. Nevertheless, the mortality rate in the presence of SH could be increased by approximately eight-fold in more hyperglycemic (BG > 250 mg/dL) than less hyperglycemic (BG < 200 mg/dL) conditions. Consequently, SH is not considered as a triggering factor of increased mortality; however, it could be manifested as a comorbid condition in the setting of aggravated underlying conditions. During the primary assessment of ill-looking children upon arrival to the PED, therefore, immediate glucose monitoring should be a critical step.

SH is caused by increased gluconeogenesis, glycogenolysis, and insulin resistance [14]. This mechanism is the result of an increase in counterregulatory hormones, including epinephrine, norepinephrine, glucagon, cortisol, growth hormone (GH), and proinflammatory cytokines such as TNF-α, IL-1 and IL-6. Pro-inflammatory cytokines stimulate gluconeogenesis in the kidney and decrease insulin secretion by beta cells through α-adrenergic receptors. Increased GH promotes alanine release from muscle to maintain hepatic gluconeogenesis. In addition, counterregulatory hormones and proinflammatory cytokines promote insulin resistance in the liver, muscle, and adipose tissue [2]. The resulting SH causes excessive glycolysis and oxidative phosphorylation, which in turn increases the production of reactive oxygen species, resulting in mitochondrial dysfunction and changes in energy metabolism. Ultimately, cellular apoptosis increases and, as a result, organ system failure occurs [14] (Figure 3). That is, in patients with acute illness, SH is generated as a metabolic response to various inflammatory stimuli such as counterregulatory hormones and pro-inflammatory cytokines, and the generated SH can also trigger various inflammatory responses, which leads to mortality. Such a phenomenon would adversely affect the life expectancy of children in ICUs, particularly those with underlying chronic diseases, including pediatric diabetes. However, previous studies have reported minimal causality between clinical outcomes and childhood SH in critical care settings [15,16].

Although SH is frequently manifested during clinical emergencies, various controversies exist since no guidelines have specifically stated how to diagnose SH and when to start managing it. The latest guidelines define hyperglycemia as a plasma glucose level of >140 mg/dL, and treatment is required if the BG level is consistently above this value [17]. Because an equivalent fasting cut-off, BG levels, or glycated hemoglobin measurements are used for the diagnosis of diabetes mellitus in both children and adults, the hyperglycemia cut-off suggested by the US guidelines seems applicable to the pediatric population [18]. Therefore, appropriate treatment should be considered for children with SH and plasma glucose levels of >150 mg/dL.

The optimal management of SH is controversial because it can be either transient or persistent. In the emergency department, the priority of treatment is determined according to the patient’s acuity, and the acuity is determined by the patient’s symptoms and initial vital signs. In addition, the purpose of treatment in the emergency department is to relieve the symptoms the patient is complaining about rather than the continuous management of the patient. Therefore, SH, which does not appear to directly affect the patient’s condition, is inevitably drawn away from the doctor’s attention. Some studies recommend prompt initiation of insulin therapy above a certain glucose level; however, there are no evidence-based data on whether approving such treatment is beneficial [19]. Instead, maintaining strict glycemic control by treating the underlying disease causing the SH could be more reasonable for patients with normal glucose metabolism. Therefore, resolving hyperglycemia is challenging if the exact underlying cause is not known or if the treatment response is poor. According to our data, the role of SH in emergency care settings appears to be valuable in predicting disease progression and prognosis. Thus, thorough assessment and prompt intervention against the underlying cause should be prioritized in the overall management of SH in children visiting the PED.

This study had several limitations. First, it was a retrospective study that extracted and analyzed data from the electronic medical records of a single institution. Therefore, it may be difficult to generalize the results for all patients who visit the emergency room. However, since we used a large data set spanning over 10 years for our analysis, the above limitation was minimized. Second, since the data of this study were provided anonymously, the possibility that the information of the same patient might be included more than once cannot be excluded. These are the inherent limitations of de-identified datasets and are not specific to this study.

## 5. Conclusions

SH is common in children with acute illness and is associated with higher hospitalization and mortality rates. Therefore, when SH is identified in children who visit the PED, there is a need to immediately identify the cause and take prompt intervention to resolve it.

## Figures and Tables

**Figure 1 children-09-00505-f001:**
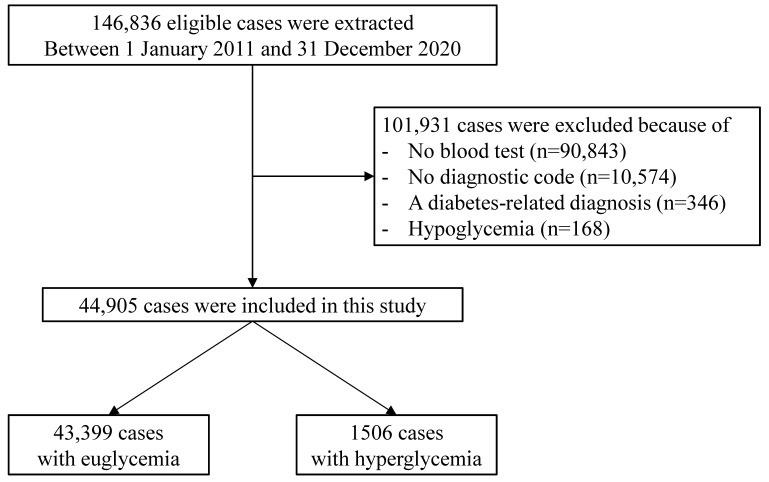
Flowchart of the study population.

**Figure 2 children-09-00505-f002:**
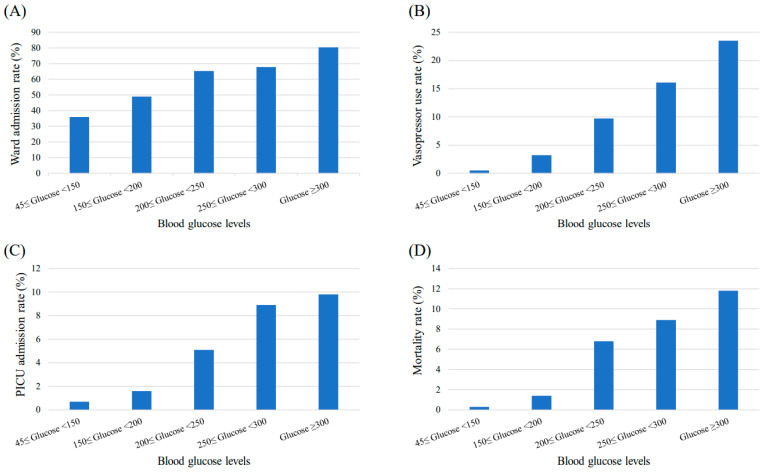
Changes in patient outcomes according to blood glucose levels. The (**A**) ward admission rate, (**B**) vasopressor use rate, (**C**) PICU admission rate, and (**D**) mortality rate of patients visiting the PED increase significantly as the blood glucose levels increase (*p* < 0.001). Abbreviations: PICU, pediatric intensive care unit; PED, pediatric emergency department.

**Figure 3 children-09-00505-f003:**
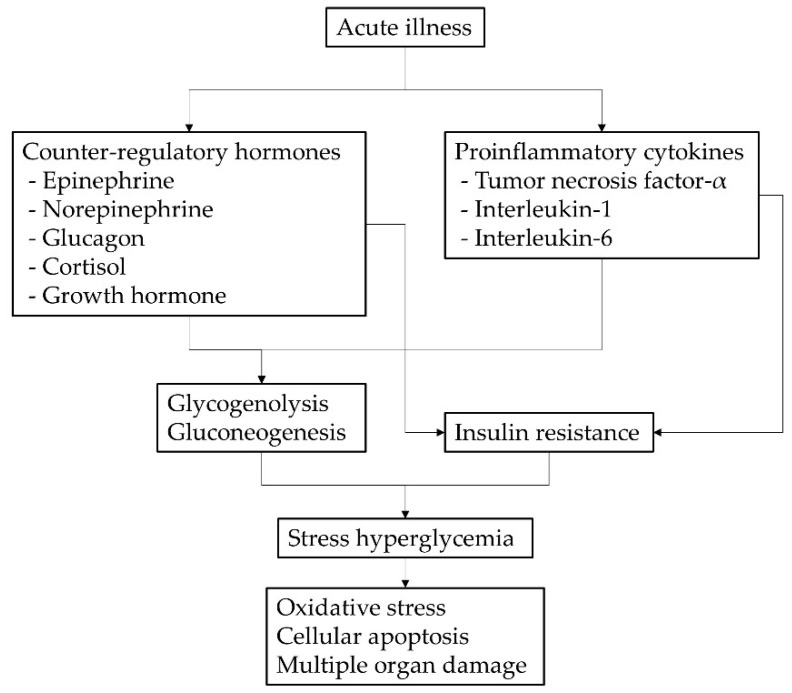
Pathophysiology of stress hyperglycemia in children with acute illness.

**Table 1 children-09-00505-t001:** Characteristics of patients visiting the pediatric emergency department with stress hyperglycemia and normoglycemia (*n* = 44,905).

Variables	Normoglycemia (*n* = 43,399)	Stress Hyperglycemia (*n* = 1506)	*p* Value
Age (month), mean ± SD	82.7 ± 68.1	74.5 ± 65.4	<0.001 ^a^
0–12 months	6146 (14.2)	180 (12.0)	<0.001 ^b^
1–3 years	9428 (21.7)	447 (29.7)	
4–6 years	8228 (19.0)	311 (20.7)	
7–12 years	9091 (20.9)	258 (17.1)	
13–17 years	10,506 (24.2)	310 (20.6)	
Female	19,452 (44.8)	620 (41.2)	0.007 ^b^
Diagnosis			
Respiratory	13,412 (30.9)	554 (36.8)	<0.001 ^b^
Gastrointestinal	14,767 (34.0)	229 (15.2)	
Hemato-oncologic	4392 (10.1)	216 (14.3)	
Neurologic	2558 (5.9)	210 (13.9)	
Trauma	944 (2.2)	74 (4.9)	
Infectious	2558 (5.9)	55 (3.7)	
Uro-genital	1196 (2.8)	21 (1.4)	
Cardiac	517 (1.2)	23 (1.5)	
Others	3055 (7.1)	124 (8.2)	
Mode of arrival			
Self-referred	32,837 (75.7)	1162 (77.2)	<0.001 ^b^
Outpatient department	3680 (8.5)	158 (10.5)	
Referred from clinic	6873 (15.8)	185 (12.3)	

All values are frequency (%) except where otherwise indicated. ^a^ *p*-value from the *t*-test. ^b^ *p*-value from the Chi-squared test.

**Table 2 children-09-00505-t002:** Comparison of outcomes for patients with stress hyperglycemia and normoglycemia (*n* = 44,905).

Variable	Normoglycemia (*n* = 43,399)	Stress Hyperglycemia (*n* = 1506)	*p*
PED Length of stay (h), mean ± SD	7.4 ± 9.0	9.2 ± 10.9	<0.001 ^a^
Hospitalization	15,577 (35.9)	792 (52.6)	<0.001 ^b^
Vasopressor administration	223 (0.5)	77 (5.1)	<0.001 ^b^
PICU admission	312 (0.7)	39 (2.6)	<0.001 ^b^
Mortality	123 (0.3)	40 (2.7)	<0.001 ^b^

All values are frequency (%) except where otherwise indicated. ^a^ *p*-value from the *t*-test. ^b^ *p*-value from the Chi-squared test. PED: pediatric emergency department, PICU: pediatric intensive care unit.

**Table 3 children-09-00505-t003:** Logistic regression analysis of patient outcomes with blood glucose levels (*n* = 44,905).

Variable	Normoglycemia (*n* = 43,399)	Stress Hyperglycemia (*n* = 1506)			
		150 ≤ Glucose < 200 (*n* = 1223)	200 ≤ Glucose < 250 (*n* = 176)	250 ≤ Glucose < 300 (*n* = 56)	Glucose ≥ 300 (*n* = 51)
Hospitalization	1 (reference)	1.77 (1.57–1.99)	3.29 (2.39–4.54)	4.06 (2.28–7.23)	7.39 (3.61–15.13)
*p*-value		<0.001	<0.001	<0.001	<0.001
vasopressor administration	1 (reference)	6.61 (4.67–9.33)	21.39 (12.73–35.92)	38.64 (18.68–79.94)	58.23 (29.27–115.85)
*p*-value		<0.001	<0.001	<0.001	<0.001
PICU admission	1 (reference)	2.59 (1.63–4.10)	7.64 (3.82–15.26)	17.11 (6.63–44.17)	10.78 (3.76–30.92)
*p*-value		<0.001	<0.001	<0.001	<0.001
Mortality	1 (reference)	5.61 (3.35–9.37)	27.96 (14.95–52.26)	44.22 (17.03–114.82)	39.94 (16.31–97.81)
*p*-value		<0.001	<0.001	<0.001	<0.001

Values are presented as odds ratio (95% confidence interval). Logistic regression analysis was adjusted for age, gender, diagnosis, and mode of arrival. PICU: pediatric intensive care unit.

## Data Availability

Data are available on request owing to restrictions, such as privacy or ethics.

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
