# Peer review of "Association between Stress Hyperglycemia and Adverse Outcomes in Children Visiting the Pediatric Emergency Department"

_children, 2022, doi:10.3390/children9040505_

Round 1

Reviewer 1 Report

The paper addresses a very interesting topic for paediatricians in emergency departments (PED) all over the world as stress hyperglycemia (SH) represents a very common disease. Moreover, it is very clearly exposed and the design is well developed in all its parts.

However, there are some points to be improved.

The introduction must include other notions from previous literature as: which are the mechanisms that lead to adverse outcomes in adults and children with SH? Are there any studies about pathophysiology? Are these events directly correlated to SH or is SH one of the manifestations of the pathologic status? What are the other adverse events besides mortality? In literature, what diseases of the patients in PED are most commonly related to SH?

In the discussion section, the paragraph (lines 146-158) on the conditions associated to SH could be moved to the introduction, as the paragraph dealing with the causes of SH (lines 182-189).

In line 167 it is stated that “mortality rate due to SH”. How can it be asserted that the mortality is due to SH and not to the underlying pathological condition? Is there any evidence in the literature to explain this?

Furthermore, in this chapter it might be interesting to discuss and better focus on the results of the study. In particular, observing the results of table 3, it is evident that the PICU admission and mortality rate decrease in the last group of patients (glucose>300) instead of increasing, compared to the 250<glucose<300 group. How do you explain these results?

And how do you explain that SH incidence is higher in 1-3 year-old patients?

Finally, it could be interesting to compare the results with those of adult series already published.

Author Response

The paper addresses a very interesting topic for paediatricians in emergency departments (PED) all over the world as stress hyperglycemia (SH) represents a very common disease. Moreover, it is very clearly exposed and the design is well developed in all its parts.

However, there are some points to be improved.

Rsponse:

We would like to thank the editor and reviewers for their thoughtful comments on the manuscript. In the manuscript, revised texts are highlighted in yellow. We hope the editor and reviewers would agree on the revised version for the publication with quality improved.

  1. The introduction must include other notions from previous literature as: which are the mechanisms that lead to adverse outcomes in adults and children with SH? Are there any studies about pathophysiology? Are these events directly correlated to SH or is SH one of the manifestations of the pathologic status? What are the other adverse events besides mortality? In literature, what diseases of the patients in PED are most commonly related to SH? In the discussion section, the paragraph (lines 146-158) on the conditions associated to SH could be moved to the introduction, as the paragraph dealing with the causes of SH (lines 182-189).

Response:

Thank you for your important note.

We agree with reviewer’s points. Based on the reviewer's opinion, the introduction section was revised by referring to previous studies on mechanisms that cause adverse outcomes in patients with SH. According to Fattorusso et al.(2018), when metabolic demand increases in patients with acute illness, blood glucose rises to provide adequate fuel to vital organs, and this elevated blood glucose is referred to as SH. This increased blood glucose persists in the chronic phase of the disease, increasing oxidative damage and enhancing the proinflammatory response, which has a harmful effect on the patient. Thus, SH is not a specific disease, but rather a comorbid condition in the setting of aggravated underlying disease.

In addition to mortality in patients with SH, other adverse outcomes include hospital length of stay (LOS) and ICU admission. According to Tsai et al.(2019), the hospital LOS in patients with SH was 16.4 days, which was significantly longer than 7.8 days in patients with normoglycemia. Also, the ICU admission rate was 55.6% in patients with SH, which was significantly higher than 20.9% in patients with normoglycemia.

According to Valerio et al.(2001), febrile seizure was the most common comorbidity in the presence of SH in children visiting the PED. This study also reported a 3-fold higher SH in children with febrile seizure compared to other febrile disorders. In addition, according to Levmore-Tamir et al.(2020), the most common diseases in children with SH at PED visits were respiratory (37.8%), neurologic (14.8%), and gastrointestinal (14.1%) diseases.

As commented by the reviewer, from the discussion section, the paragraphs about states related to SH (lines 146-158) and the paragraphs dealing with the causes of SH (lines 182-189) have been moved to the introduction section. By doing so, I think that the content of this study will be easy to understand for the readers.

We rephrased the text as the following to support the ideas above:

Formerly in the manuscript (Line 29):

Acute illness increases cortisol, glucagon, growth hormone, catecholamine, and various cytokines in the body, which stimulate glycogenolysis and gluconeogenesis, resulting in SH. In addition, glucose intolerance, peripheral insulin resistance, and restriction of urine glucose excretion due to dehydration may also act as triggers.

Contents rephrased in revised manuscript (Line 29):

There are numerous conditions associated with SH in children who visit the pediatric emergency room. According to Levmore-Tamir et al.[4], the most common diseases in children with SH at pediatric emergency department (PED) visits were respiratory, neurologic, and gastrointestinal diseases. In addition, according to Valerio et al. [5], febrile seizure was the most common comorbidity in the presence of SH in children visiting the PED. This study also reported a 3-fold higher SH in children with febrile seizure compared to other febrile disorders. Furthermore, upon a child’s visit to the PED, it is critical to promptly administer certain drugs, such as inhaled beta-adrenergic stimulants, glucocorticoid, intravenous dextrose fluid, phenytoin, and antipyretics (acetaminophen or ibuprofen), which at the same time, could lead to drug-induced hyperglycemia [6].

In patients with acute illness, when the metabolic demand increases, blood glucose rises in order to supply adequate fuel to vital organs, and this elevated blood glucose is called SH. The most common mechanism of SH is explained by increased gluconeogenesis and glycogenolysis, as well as insulin resistance stimulated by proinflammatory cytokines and counter-regulatory hormones secreted during critically ill conditions. This then leads to glycolysis and oxidative phosphorylation, resulting in excessive reactive oxygen species, cellular apoptosis, and multiple organ damage [2].

We added the text as the following:

Contents rephrased in revised manuscript (Line 52):

In addition, the hospital LOS in patients with SH was 16.4 days, which was significantly longer than 7.8 days in patients with normoglycemia. Also, the ICU admission rate was 55.6% in patients with SH, which was significantly higher than 20.9% in patients with normoglycemia [7].

  1. In line 167 it is stated that “mortality rate due to SH”. How can it be asserted that the mortality is due to SH and not to the underlying pathological condition? Is there any evidence in the literature to explain this? Furthermore, in this chapter it might be interesting to discuss and better focus on the results of the study. In particular, observing the results of table 3, it is evident that the PICU admission and mortality rate decrease in the last group of patients (glucose>300) instead of increasing, compared to the 250<glucose<300 group. How do you explain these results?

Response:

Thank you for pointing out the causality between SH and mortality rate.

We agree with the reviewer’s point that SH by itself cannot simply increase the mortality rate. As mentioned in line 164-9, however, our results showed SH was clearly correlated with poor outcome measures, thus SH could be manifested as a comorbid condition for those with poor prognosis. Furthermore, it seemed SH could worsen in the setting of aggravated underlying condition. As our recommendations on line 170-2 and 198-9, the assessment of SH could be an important action to take during the emergent assessment of critically ill pediatric patients.

We also agree with the reviewer’s comment regarding lower odds ratios for PICU admission and mortality revealed in the last group (glucose>300) compared to those of 250<glucose<300 group. Firstly, there is a statistical drawback due to the number of subjects included in each category among stress hyperglycemic patients. Compared to a large number of normoglycemic patients, number of patients determined as SH category decreases as blood glucose level increases. For example, compared to n=43,399 of normoglycemia group, the last group only consists of 51 patients. If the sample size is small like the last group, it is considered an error that occurs because the group does not sufficiently reflect the adjusted variables when statistically adjusted. Secondly, for those who expired in the emergency department were excluded for statistical analysis for PICU admission since they never had admitted to PICU. If they had expired in PICU not in emergency department, the odd ratio for PICU admission might have been the highest among SH groups.

We rephrased the text as the following to support the ideas above:

Formerly in the manuscript (Line 166):

In addition, all outcome parameters progressively increased as the BG levels increased, implying that the mortality rate due to SH increase by approximately eight-fold in more hyperglycemic (BG >250 mg/dL) than less hyperglycemic (BG <200 mg/dL) conditions. Our findings also suggest that children with extreme hyperglycemia are expected to have poorer outcomes; therefore, immediate glucose monitoring should be a critical step during the primary assessment of ill-looking children upon arrival to the PED.

Contents rephrased in revised manuscript (Line 174):

In Figure 2, all outcome parameters had tendency to increase as the BG levels increased. Likewise, all adjusted odds ratios for the patient outcomes appeared to be the highest in extremely hyperglycemic (BG>300 mg/dL) group except for PICU admission and mortality (Table 3). Possible explanations are i) statistical limitations due to the small sample size of the extremely hyperglycemic group ii) exclusion for those numbers who expired in the PED unit could have affected the odds ratio for PICU admission. Nevertheless, the mortality rate in the presence of SH could be increased by approximately eight-fold in more hyperglycemic (BG >250 mg/dL) than less hyperglycemic (BG <200 mg/dL) conditions.  Consequently, SH is not considered as a triggering factor of increased mortality; however, it could be manifested as a comorbid condition in the setting of aggravated underlying condition. During the primary assessment of ill-looking children upon arrival to the PED, therefore, immediate glucose monitoring should be a critical step.

  1. How do you explain that SH incidence is higher in 1-3 year-old patients?

Response:

Thank you for another valuable pointing out.

As mentioned in the manuscript, children aged 13-17 years accounted for the largest proportion of entire patients who visited the PED (Line 160). Among those with SH, however, the largest proportion was children aged 1-3 years. Children aged 1-3 years accounted for the second largest proportion among the entire PED visitors.

In the present study, we subcategorized patients based on age and disease type, but not based on chief complaint. Most common chief complaint of children aged 1-3 years upon PED arrival was febrile illness due to any of respiratory, gastrointestinal, hemato-oncologic, neurologic, infectious, urogenital, or cardiac origin. As mentioned in line 148, childhood SH appears to be commonly present in febrile condition. SH could trigger inflammatory cascade, however at the same time, it could be a certain reaction triggered by inflammation.

We rephrased the text as the following to support the ideas above:

Formerly in the manuscript (Line 159):

We also found that SH incidence was higher in children aged between 1-3 years compared to other age groups, although those who are 13-17 years of age accounted for the largest proportion (24.1%) of patients who visited the PED.

Contents rephrased in revised manuscript (Line 164):

Interestingly, the frequency for children aged 1-3 years was the highest in SH population compared to other age groups, and the second highest among the entire PED visitors. In our study, patients were subcategorized by age and disease type, not by chief complaint, and the most common reason for PED visit among children aged 1-3 years was febrile illness. Higher frequency of febrile population might have affected the SH incidence of children aged 1-3 years.

  1. Finally, it could be interesting to compare the results with those of adult series already published.

Response:

We certainly agree with reviewer’s suggestion that it is important to introduce relevant results from the adult studies.

In a Turkish study, the mortality rate was approximately 10 times higher in patients diagnosed with acute myocardial infarction with SH compared to those without SH. (DOI: 10.5114/amsad.2019.87303). Another study from Italy reported that high stress hyperglycemia ratio among all outcome predictors increased the risk of mortality by over 5 times in elderly diabetic patients following diagnosis of sepsis-related hospitalization. (DOI: 10.1097/CCE.0000000000000152)

We added the text as the following:

Contents added in revised manuscript (Line 154):

In terms of mortality rate in particular, our results were consistent with findings from previous studies conducted in adult population. Cinar et al reported the mortality rate among patients diagnosed with acute myocardial infarction was approximately 10 times higher in SH group than non-SH group [12]. In another study by Fabbri et al, high stress hyperglycemia ratio among all outcome predictors increased the risk of mortality by over 5 times in elderly diabetic patients following diagnosis of sepsis-related hospitalization [13]. Likewise, the mortality rate in our results revealed 44-fold increase in SH population compared to normoglycemic population.

Reviewer 2 Report

Dear Authors,

I read with interest your paper entitled “Association between stress hyperglycemia and adverse outcomes in children visiting the pediatric emergency department”. The topic is relevant and interesting.

I think however the discussion should be re-formulated in a better order, specifying first of all where the novelty of your study lies, then why is SH a by-product of stress, why is this a protective mechanism in the “idea” of physiology and why it then deranges to hyperglycemia in certain situations; and then only after why it can be detrimental to other organs, what the implications for therapy are, why in your opinion there is no definite guideline to treatment in the ED (very interesting point)

Page 2 line 50: I think you need to add the word “who” between “and “ and “were diagnosed”

Page 2 line 94: I would substitute “women” with  “female sex” since we are talking of children

Page 5 line 126 to 131: I would not write this info in the text since you stated in the following table the same results and it is difficult to read. I would only summarize that ORs for the defined outcomes are summarized in the table

Page 5 line 146: I think it would be important to add here the discussion on the genesis of SH as product of glycogenolysis and gluconeogenesis as effect of pro-inflammatory cytokines and stress hormones, and expand it a little bit. It would be also nice to put this into a schematic figure explaining these mechanisms so that it is a good reminder of physiology

Then, only after, you can say it also true the opposite, that SH stimulates stress hormones and…

Page 6 line 156: I think you should divide more clearly this concept: Sh is a product of a various stimuli AND can act as a trigger to cytokine production – these are two separate concepts and in one SH is a product, in the other is an effect

Why would you say the hyperglycemia is higher in smaller children? What do you think potential mechanisms are?

Author Response

I read with interest your paper entitled “Association between stress hyperglycemia and adverse outcomes in children visiting the pediatric emergency department”. The topic is relevant and interesting.

Response:

Thank you for taking the time to review our manuscript and for providing meaningful comments. We have made corrections and clarifications in the revised manuscript after carefully reading your comments.

  1. I think however the discussion should be re-formulated in a better order, specifying first of all where the novelty of your study lies, then why is SH a by-product of stress, why is this a protective mechanism in the “idea” of physiology and why it then deranges to hyperglycemia in certain situations; and then only after why it can be detrimental to other organs, what the implications for therapy are, why in your opinion there is no definite guideline to treatment in the ED (very interesting point)

Response: 

Thank you for your important comment. We have revised the discussion section to reflect your comments.

We have revised the discussion section to reflect your comments. In the first paragraph, the novelty of this study was introduced by comparing it with previous studies. The second and third paragraphs describe the interpretation of the results of this study. The fourth paragraph describes the causes and pathophysiology of SH, and the effects of SH on the disease and condition of patients. In the fifth paragraph, treatment criteria for SH are presented. In the next paragraph, the reasons for the lack of clear treatment guidelines for SH in children visiting the PED and management guidelines of SH are presented.

We re-formulate the entire discussion section, and the content is too large to be attached to this document. We attached only the newly added text as the following:

Contents added in revised manuscript (Line 215)

In the emergency department, the priority of treatment is determined according to the patient's acuity, and the acuity is determined by the patient's symptoms and initial vital signs. In addition, the purpose of treatment in the emergency department is to relieve the symptoms the patient is complaining about rather than the continuous management of the patient. Therefore, SH, which does not appear to directly affect the patient's condition, is inevitably drawn away from the doctor's attention.

  1. Page 2 line 50: I think you need to add the word “who” between “and “ and “were diagnosed”

Response:

Thank you for pointing this out. We made the corrections as the reviewer pointed out.

Contents rephrased in revised manuscript (Line 65)

Patients without blood tests, not assigned with a diagnostic code, and who were diag-nosed with diabetes or hypoglycemia were excluded from this study.

  1. Page 2 line 94: I would substitute “women” with  “female sex” since we are talking of children

Response:

Thank you for your accurate point. I was mistaken. Correctly changed to “female sex” instead of “women”.

Contents rephrased in revised manuscript (Line 109)

The proportion of female sex in the SH group was also significantly lower (44.8% vs. 41.2%, P<0.001).

  1. Page 5 line 126 to 131: I would not write this info in the text since you stated in the following table the same results and it is difficult to read. I would only summarize that ORs for the defined outcomes are summarized in the table

Response:

Thank you for your important note. I agree with your opinion that it is better to organize the content as concise and simple as possible, considering the readability of the readers. I will reduce what is already in the table.

We rephrased the text as the following to support the ideas above:

Formerly in the manuscript (Line 126):

Compared to the normoglycemic group, the ORs (95% CI) for ward admission rates were 1.77 (1.57–1.99), 3.29 (2.39–4.54), 4.06 (2.28–7.23), and 7.39 (3.61–15.13) for levels 150-<200, 200-<250, 250-<300, and ≥300 mg/dL, respectively. Meanwhile, the ORs (95% CI) for PICU admission rates were 2.59 (1.63–4.10), 7.64 (3.82–15.26), 17.11 (6.63–44.17), and 10.78 (3.76–30.92) for each BG level, respectively. Lastly, the mortality w 5.61 (3.35–9.37), 27.96 (14.95–52.26), 44.22 (17.03–114.82), and 39.94 (16.31–97.81) for each group, respec-tively (Table 3).

Contents rephrased in revised manuscript (Line 141):

Compared with the normoglycemic group, the OR (95% CI) for adverse outcomes including hospitalization, vasopressor administration, PICU admission, and mortality for each group of SH are specified in Table 3.

  1. Page 5 line 146: I think it would be important to add here the discussion on the genesis of SH as product of glycogenolysis and gluconeogenesis as effect of pro-inflammatory cytokines and stress hormones, and expand it a little bit. It would be also nice to put this into a schematic figure explaining these mechanisms so that it is a good reminder of physiology

Then, only after, you can say it also true the opposite, that SH stimulates stress hormones and…

Response:

We certainly agree with the reviewer's suggestion that it is important to add more about the genesis of SH. In addition, a schematic figure explaining the mechanism of SH generation has been added.

We rephrased the text and added schematic figure as the following:

Formerly in the manuscript (Line 152-154):

SH also induces the activity of the sympathetic system, triggering the hypersecretion of counterregulatory hormones, such as epinephrine, cortisol, glucagon, growth hormones, and pro-inflammatory cytokines, such as tumor necrosis factor-α [16].

Contents rephrased in revised manuscript (Line 186-193):

SH is caused by increased gluconeogenesis, glycogenolysis, and insulin resistance [14]. This mechanism is the result of an increase in counterregulatory hormones including epinephrine, norepinephrine, glucagon, cortisol, growth hormone(GH), and proinflammatory cytokines such as TNF-α, IL-1, and IL-6. Pro-inflammatory cytokines stimulate gluconeogenesis in the kidney and decrease insulin secretion by beta cells through α-adrenergic receptors. Increased GH promotes alanine release from muscle to maintain hepatic gluconeogenesis. In addition, counterregulatory hormones and proinflammatory cytokines promote insulin resistance in the liver, muscle, and adipose tissue [2].

  1. Page 6 line 156: I think you should divide more clearly this concept: Sh is a product of a various stimuli AND can act as a trigger to cytokine production – these are two separate concepts and in one SH is a product, in the other is an effect

Response:

Thank you for another valuable pointing out. We fully agree with your opinions. We need to clearly divide the concepts of the mechanism by which SH occurs and the effects of SH, according to the reviewer's opinion. In this study, the parts describing the mechanism of SH and its effects are divided into Line 152-158 and Line 182-189, so it seems that the contents were not conveyed effectively. The two parts are summarized in one paragraph, and the mechanism for the occurrence of SH and its effects are described separately.

We rephrased the text as the following:

Formerly in the manuscript (Line 152-158):

SH also induces the activity of the sympathetic system, triggering the hypersecretion of counterregulatory hormones, such as epinephrine, cortisol, glucagon, growth hormones, and pro-inflammatory cytokines, such as tumor necrosis factor-α [16]. Therefore, SH can exacerbate inflammation, induce oxidative stress, and worsen disease prognosis. This al-so suggests that SH may be a metabolic response to various inflammatory stimuli, as it can trigger proinflammatory cytokine signaling, which leads to mortality.

Formerly in the manuscript (Line 182-189):

The most common mechanism of SH is explained by increased gluconeogenesis and glycogenolysis, as well as insulin resistance stimulated by proinflammatory cytokines and counter-regulatory hormones secreted during critically ill conditions. This then leads to glycolysis and oxidative phosphorylation, resulting in excessive reactive oxygen species, cellular apoptosis, and multiple organ damage [19]. Such a phenomenon would adversely affect the life expectancy of children in ICUs, particularly those with underlying chronic diseases, including pediatric diabetes. But previous studies have reported minimal cau-sality between clinical outcomes and childhood SH in critical care settings [20,21].

Contents rephrased in revised manuscript (Line 186-200):

SH is caused by increased gluconeogenesis, glycogenolysis, and insulin resistance [14]. This mechanism is the result of an increase in counterregulatory hormones including epinephrine, norepinephrine, glucagon, cortisol, growth hormone(GH), and proinflammatory cytokines such as TNF-α, IL-1, and IL-6. Pro-inflammatory cytokines stimulate gluconeogenesis in the kidney and decrease insulin secretion by beta cells through α-adrenergic receptors. Increased GH promotes alanine release from muscle to maintain hepatic gluconeogenesis. In addition, counterregulatory hormones and proinflammatory cytokines promote insulin resistance in the liver, muscle, and adipose tissue [2]. The resulting SH causes excessive glycolysis and oxidative phosphorylation, which in turn increases the production of reactive oxygen species, resulting in mitochondrial dysfunction and changes in energy metabolism. Ultimately, cellular apoptosis increases and, as a result, organ system failure occurs. [14]. (Figure 3) That is, in patients with acute illness, SH is generated as a metabolic response to various inflammatory stimuli such as counterregulatory hormones and pro-inflammatory cytokines, and the generated SH can also trigger various inflammatory responses, which leads to mortality.

  1. Why would you say the hyperglycemia is higher in smaller children? What do you think potential mechanisms are?

Response:

Thank you for another valuable pointing out.

As mentioned in the manuscript, children aged 13-17 years accounted for the largest proportion of entire patients who visited the PED (Line 160). Among those with SH, however, the largest proportion was children aged 1-3 years. Children aged 1-3 years accounted for the second largest proportion among the entire PED visitors.

In the present study, we subcategorized patients based on age and disease type, but not based on chief complaint. Most common chief complaint of children aged 1-3 years upon PED arrival was febrile illness due to any of respiratory, gastrointestinal, hemato-oncologic, neurologic, infectious, urogenital, or cardiac origin. As mentioned in line 148, childhood SH appears to be commonly present in febrile condition. SH could trigger inflammatory cascade, however at the same time, it could be a certain reaction triggered by inflammation.

We rephrased the text as the following to support the ideas above:

Formerly in the manuscript (Line 159):

We also found that SH incidence was higher in children aged between 1-3 years compared to other age groups, although those who are 13-17 years of age accounted for the largest proportion (24.1%) of patients who visited the PED.

Contents rephrased in revised manuscript (Line 164-169):

Interestingly, the frequency for children aged 1-3 years was the highest in SH population compared to other age groups, and the second highest among the entire PED visitors. In our study, patients were subcategorized by age and disease type, not by chief complaint, and the most common reason for PED visit among children aged 1-3 years was febrile illness. Higher frequency of febrile population might have affected the SH incidence of children aged 1-3 years.

Round 2

Reviewer 2 Report

No more comments for me, thanks